# Assessing Cellular Uptake of Exogenous Coenzyme Q_10_ into Human Skin Cells by X-ray Fluorescence Imaging

**DOI:** 10.3390/antiox11081532

**Published:** 2022-08-06

**Authors:** Theresa Staufer, Mirja L. Schulze, Oliver Schmutzler, Christian Körnig, Vivienne Welge, Thorsten Burkhardt, Jens-Peter Vietzke, Alexandra Vogelsang, Julia M. Weise, Thomas Blatt, Oliver Dabrowski, Gerald Falkenberg, Dennis Brückner, Carlos Sanchez-Cano, Florian Grüner

**Affiliations:** 1Universität Hamburg and Center for Free-Electron Laser Science (CFEL), Institute for Experimental Physics, Faculty for Mathematics, Informatics and Natural Sciences, Luruper Chaussee 149, 22761 Hamburg, Germany; 2Research and Development, Beiersdorf AG, Unnastrasse 48, 20245 Hamburg, Germany; 3Fraunhofer Institute for Applied Polymer Research (IAP), Center for Applied Nanotechnology (CAN), Grindelallee 117, 20146 Hamburg, Germany; 4Deutsches Elektronen-Synchrotron DESY, Notkestrasse 85, 22607 Hamburg, Germany; 5DIPC, Paseo Manuel de Lardizabal 4, 20018 Donostia-San Sebastian, Spain; 6Ikerbasque, Basque Foundation for Science, Plaza de Euskadi 5, 48009 Bilbao, Spain

**Keywords:** X-ray fluorescence (XRF) imaging, Q_10_, uptake, single skin cells

## Abstract

X-ray fluorescence (XRF) imaging is a highly sensitive non-invasive imaging method for detection of small element quantities in objects, from human-sized scales down to single-cell organelles, using various X-ray beam sizes. Our aim was to investigate the cellular uptake and distribution of Q_10_, a highly conserved coenzyme with antioxidant and bioenergetic properties. Q_10_ was labeled with iodine (I_2_-Q_10_) and individual primary human skin cells were scanned with nano-focused beams. Distribution of I_2_-Q_10_ molecules taken up inside the screened individual skin cells was measured, with a clear correlation between individual Q_10_ uptake and cell size. Experiments revealed that labeling Q_10_ with iodine causes no artificial side effects as a result of the labeling procedure itself, and thus is a perfect means of investigating bioavailability and distribution of Q_10_ in cells. In summary, individual cellular Q_10_ uptake was demonstrated by XRF, opening the path towards Q_10_ multi-scale tracking for biodistribution studies.

## 1. Introduction

A detailed understanding of the biodistribution of entities such as medical drug compounds, immune cells or antibodies is essential in life sciences. Molecular imaging modalities contribute to a profound understanding of biological processes at the full body, cellular, and even subcellular level. At the same time, imaging technologies can reveal information on the spatial and temporal biodistribution of molecules, and thus, an idea of the biological response to be expected [1]. In order to visualize and quantitatively determine uptake and localization of structural or chemical components into specific organs or cells, a non-invasive method offering both high resolution and sensitivity is advantageous. X-ray fluorescence (XRF) imaging meets all the major requirements for highly scientific evaluations and insights into specific modes of action: high spatial resolution (down to the nanometer scale), high sensitivity (detection of elements down to fg/cell-levels), as well as temporal resolution. The latter is of the highest relevance, because the markers applied do not decay over time, such as in single-photon emission computed tomography (SPECT) imaging or positron emission tomography (PET) [2,3,4,5,6,7,8,9,10,11].

PET/SPECT are inappropriate for single-cell assessments with sub-cellular spatial resolution, and in contrast with the commonly used method of inductively coupled plasma-mass spectrometry (ICP-MS) [12], XRF is non-destructive and thus well suited even for in vivo longitudinal studies [2,3,4,5,6,7,8,9]. Finally, dedicated spatial filtering enables the possibility of overcoming the major challenge of intrinsic multiple Compton scattering when translating XRF from in vitro to large study objects, such as human organs or humans [4,8].

In order to make dedicated entities of interest, e.g., medical drug compounds or biomarkers, detectable via XRF, they need to be labeled with either molecular contrast agents or nanoparticles. The choice of the marker requires profound consideration, and depends on several factors, such as the synthetic processes required for the labeling, object size and incident X-ray energy, which has to be higher than the absorption edge of the used element. Furthermore, one has to ensure that naturally occurring elements in the scanned sample do not show an overlap of signals and, of course, marker elements have to be non-toxic and not interfere with or influence biological processes.

Coenzyme Q_10_ (Q_10_), also referred to as ubiquinone, is an essential and highly conserved cellular component mainly present in cellular and mitochondrial membranes, where it acts as a cofactor of mitochondrial respiratory chain complexes supporting cellular bioenergetics in the process of oxidative phosphorylation [13]. In its reduced form (known as ubiquinol), it exerts antioxidant activities. The skin, which is exposed to the external environment like no other organ, is confronted by damaging pro-oxidant stimuli, ultraviolet radiation, and pollution, which are all triggers of cellular oxidative stress and senescence processes. Consequences are skin ageing features such as loss of skin’s firmness and elasticity, wrinkle formation, and susceptibility to inflammatory processes which accelerate the ageing phenotype outcome [14].

With increasing age, Q_10_ content decreases [15] resulting in significant inhibition of the mitochondrial respiration and leading to increased oxidative stress [16,17,18]. For instance, exogenous Q_10_ supplementation significantly increased respiration parameters in ex vivo human epidermis and antioxidant capacity in vitro and in vivo, and also led to improved features of aged skin in healthy human subjects [15,17,19,20]. Skin Q_10_ supplementation strategies have been accepted as anti-ageing approaches [20].

As Q_10_ is an essential cellular component and is highly conserved across tissues and species, also other organs suffer from an age-, but also disease-dependent decrease in Q_10_ content. In fact, for conditions associated both with mitochondrial dysfunction and oxidative stress, such as degenerative pathologies and ageing-relevant physiological conditions, a decrease in Q_10_ content was detected with organ-specific differences [21] both in animal models [22] and in humans [23,24]. Notably, a beneficial impact for Q_10_ supplementation was shown in evidence-based studies for several of these conditions, such as Parkinson’s disease, Alzheimer’s disease, and heart conditions such as heart failure [25,26,27,28], demonstrating the impact of Q_10_ in proper function of organisms.

Biodistribution studies using imaging mass spectrometry [29] and PET [30] have been used to reveal the localization of Q_10_ in the mouse brain as well as the kinetics of commercially available Q_10_ supplements. In order to further understand Q_10_ uptake on a cellular level, XRF represents an ideal modality due to its high spatial resolution and the quantitative nature of the modality.

In this study, we reveal two findings:(a)The feasibility of labeling a relatively small biomolecule, i.e., Q_10_.(b)Cellular uptake and distribution of Q_10_ by means of XRF imaging.

To the best of our knowledge, no detailed quantitative uptake studies exist in the scientific literature. Our new findings obviously can pave the way towards a more detailed understanding of Q_10_ supplementation in human skin cells and, in addition, provide insights into the mode of action underlying its beneficial effects following topical treatment.

## 2. Materials and Methods

### 2.1. Analytical Pre-Assessment of I_2_-Q_10_

As iodine-labeled Q_10_ (I_2_-Q_10_) has not been described in the literature yet, it had to be newly synthesized in a multistep synthesis starting from standard Q_10_ (Kaneka Pharma). The identity of the product was confirmed by ^1^H, ^13^C and 2D NMR-analysis, as well as mass spectrometry. The chemical shift of the prominent methine group (marked with an asterisk in the molecular structure shown in Figure 1) next to the introduced iodine moiety was found to be 6.90 ppm (^1^H-NMR) and 153.34 ppm (^13^C-NMR). This is in very good accordance with similar substructures described in the literature [31]. The position of all other signals, e.g., from the benzoquinone and the isoprene units, remained unchanged. Quantitative NMR assigned the purity to 83% (Ref: Tecnazene, (1,2,4,5 Tetrachlor-3-nitrobenzene)).

The chemical structure and purity of the I_2_-Q_10_ was assessed by Nuclear Magnetic Resonance (NMR) measurements, revealing the following analytical data:

^1^H-NMR (CDCl_3_, 400 MHz): *δ* [ppm] = 6.90 (t, ^3^*J* = 6.4 Hz, 1H, C*H* = CI_2_), 5.23–4.87 (m, 9H, CH2-C*H* = C), 3.99 (s, 3H, O*Me*), 3.98 (s, 3H, O*Me*), 3.18 (d, 2H, ^3^*J* = 7.0 Hz, Ar-C*H_2_*), 2.52–1.83 (m, 39H, C-*CH_2_-CH_2_*-CH & Ar-C*H_3_*), 1.78–1.17 (m, 27H, C = C (C*H_3_*)-CH_2_). ^13^C-NMR (CDCl_3_, 100 MHz): *δ* [ppm] = 153.34 (assigned by 2D) *C*H = I_2_, 142.49* CH_2_-*C*H = C, 118.98 CH_2_-*C*H = C, 61.28 O*Me*, 39.56*, 26.59*, 25.30 Ar-*C*H_2_, 15.90* C = C(*C*H_3_)-CH_2_.

*Broad signal

HRMS (ESI): m/z calc. [C57H84I2O4 + H]^+^ = 1087.4537, found: 1087.4509 [M + H]^+^, 1109.4335 [M + Na]^+^, 1125.4142 [M + K]^+^.

Instruments: NMR: Bruker Avance I 400 MHz (Bruker, Billerica, MA, USA), HRMS: Agilent 6224 ESI-TOF, coupled Agilent HPLC 1200 Series (Agilent, Santa Clara, CA, USA).

### 2.2. Cell Culture

Primary human keratinocytes were purchased from Alphenyx (Marseille, France) and processed as described in [32]. Cells cultured at passage 2-5 were used for the experiments and cultured in EpiLife (ThermoFisher Scientific, Waltham, MA, USA) supplemented with human keratinocytes growth supplements (HKGS; ThermoFisher Scientific, Waltham, MA, USA), 1% penicillin streptomycin, and maintained in standard culture conditions at 37 °C and 5% CO_2_. For cell culture maintenance, the medium was refreshed every 2–3 days and cells were passaged or used for experiments at 80% confluence.

Q_10_ and I_2_-Q_10_ were solubilized in water using a mixture of glycerol and the emulsifying agent PEG-hydrogenated castor oil (EUMULGIN CO 40; BASF, Ludwigshafen, Germany) (Q_10_/I_2_-Q_10_:glycerol:HCO = 0.4:0.6:1) [33].

### 2.3. Cell Vitality

Briefly, cells were treated with 10, 25 or 50 µM I_2_-Q_10_ for 24 h before 3-(4,5-dimethylthiazol-2-yl)-2,5-diphenyltetrazolium bromide (MTT; Sigma Cat# M2003) or fluorescein diacetate (FDA; Sigma Cat#F7378) assays were conducted. The MTT assay was carried out according to the manufacturer’s instructions, except the plate was left on a shaker for 5 min instead of incubating overnight. Additionally, the FDA assay was conducted according to the manufacturer’s instructions. Blank values were subtracted and data normalized to untreated medium control.

### 2.4. Bioavailability

Cellular (I_2_-) Q_10_ content was quantified as previously described [34]. Briefly, cell suspensions from keratinocytes treated with 50 µM (I_2_-) Q_10_ for 24 h were extracted with isopropanol. Centrifuged isopropanol extracts were analyzed by means of LC-MS/MS (Agilent 1200 HPLC coupled to an Agilent 6490 triple quadrupole mass spectrometer; separation column: YMC-Pack Pro C8 (YMC-Group)) using adjusted multi-reaction transitions and external standard calibration. Due to the low quantity of I_2_-Q_10_, its content was measured for six technical replicates of one donor; for native unlabeled standard Q_10_ n = 5 donors, each n = 1 technical replicate was analyzed. The (I_2_-) Q_10_ content was normalized to total protein content; its content mean value is expressed as ng Q_10_/mg total protein content. For native unlabeled standard Q_10_, endogenous Q_10_ content (solvent control) was subtracted.

### 2.5. Cell Preparation for XRF Measurements

For cell pellet measurements, cells were treated with 50 µM I_2_-Q_10_ for 24 h. After treatment, cells were washed with 1× PBS twice, trypsinized and 1× PBS washed again before 400,000–750,000 cells were transferred to glass capillaries (1.0 × 80 × 0.01 mm; WJM-Glas/Müller GmbH, Berlin, Germany). Cells were centrifuged (12,000 rpm, 5 min) and kept on ice until measurement.

For single-cell measurements, silicon nitride membrane windows (Silson Ltd., Warwickshire, UK) were transferred to 6-well plates with their flat (non-etched) side up. Membranes were treated with UV-light for 1 h, before 10 µL collagen (90 µg/mL; collagen, type I solution from rat tail; Sigma-Aldrich, Saint Louis, MO, USA) was added onto the membranes and left in the incubator to dry for 1 h. Residual coating was removed by a pipette before 3,500 cells were pipetted onto each membrane. Cells were left in the incubator for adherence for 3 h before medium was added to the wells. On the next day, treatment was started by adding 50 µM I_2_-Q_10_ to the cells. After 24 h treatment incubation, cells were chemically fixated. For this, membranes were washed with 1x D-PBS before 4% paraformaldehyde was added. After 20 min incubation at room temperature, membranes were washed with 1x D-PBS and rinsed in Tris-glucose buffer (261 mM glucose, 9 mM acetic acid, 10 mM Tris buffer, pH 7.4) twice. Membranes were air dried and kept in a desiccator until measurement.

### 2.6. XRF Cell Uptake and Single Cell Measurements

#### Experimental Setup

The experiments were performed at the beamlines P21.1 and P06 at the PETRA III synchrotron at DESY, Hamburg, Germany. At the P21.1 beamline, an incident photon energy of 53.1 keV was used in a 1 mm × 1 mm beam of about 10^11^ photons/s in order to excite iodine K X-ray fluorescence lines which do not suffer from line overlap with other elements. The flux at this beamline is measured with silicon PIN diodes before and after the target area, which allows continuous monitoring.

Cell pellets after different treatments and substances were filled into glass capillaries of 1 mm diameter and 0.01 mm wall thickness and mounted on a custom-built holder to allow scanning of the samples through the X-ray beam at the P21.1 beamline. Silicon drift detectors (X-123FASTSDD and XR100FASTSDD, Amptek Inc., Bedford, MA, USA) of 17/50 mm^2^ collimated area and 0.5/1.0 mm sensor thickness were used to record the emitted characteristic X-ray fluorescence photons. The choice of 53.1 keV incident energy has the advantage that only a small number of background photons arising from Compton scattering can reach the iodine Kα line energy at around 28.6 keV. In addition, the used 0.5/1.0 mm thick SDD does only absorb a small fraction of Compton scattered photons due to the low efficiency at 45 keV, which in turn minimizes scan times by reducing the total count rate.

For single-cell micro-XRF element mapping, thin silicon nitride membrane windows (Silson Ltd., Warwickshire, England) with I_2_-Q_10_-treated cells were mounted on a custom-made holder and positioned normal to the incident beam at the P06 beamline at the PETRA III synchrotron at DESY, Hamburg, Germany [35]. In total, 3 different incident energies were used to determine the optimum parameters in terms of fluorescence cross-section and background contributions. Experiments at 35 and 40 keV incident energy with CRL lenses and CRL pre-focusing were less successful due to their comparatively low flux and high Compton scatter background. Excitation at 12 keV showed much higher sensitivity and higher spatial resolution, since KB mirrors with and without CRL pre-focusing could be used. The smallest beam size of 330 × 240 nm^2^ with 1.6 × 10^10^ photons/s was achieved at 12 keV by using KB mirrors without CRL pre-focusing. With pre-focusing, the flux could be increased up to 6 × 10^11^ photons/s on the expense of a beam size increase to 2.4 × 2.4 μm^2^. The different beam sizes allow for coarse and fine scans that are needed to determine the cell-to-cell variation in terms of uptake, but also the intracellular distribution. The iodine concentration was high enough to separate iodine L lines from overlapping calcium K lines for a dwell time per scan pixel of 1 s.

A Vortex-EM silicon drift X-ray detector of 50 mm^2^ active area and 2 mm chip thickness with an Al collimator was positioned at 10 mm distance and 80° angle to the membrane normal. The scanning positions were determined by an online microscope with 2 μm resolution installed at the beamline.

### 2.7. Data Acquisition and Analysis

#### 2.7.1. Fit Routine and Calibration in Cell Pellet Measurements

XFI is based on the excitation of tracers inside objects by high energy X-rays. The tracers thereby emit element-characteristic X-rays into the full solid angle 4π, which are detected by energy-discriminating detectors. The incident photon beam with area *A* and intensity $I_{0}$ gets attenuated by the target material before interaction with the tracer. The probability of an X-ray fluorescence photon being produced is described by the fluorescence cross-section σ. Emitted fluorescence photons again need to traverse parts of the surrounding target material and a certain distance through air before hitting the detector. Along this path, they are attenuated due to scattering or absorption processes; hence, only a fraction $T\left(E_{fluo}\right)$ of the photons with energy $E_{fluo}$ reach the detector chip. Therefore, the detected number of fluorescence photons *S* needs to be corrected for those effects.

Even if multiple detectors are used, only a fraction of the full solid angle 4π,${\;\Omega }_{det},\;$is covered, which needs to be addressed by the correction factor $\frac{4\pi }{\Omega_{\det\;\;}}$, as the fluorescence photons are emitted isotropically. The detector chip itself has limited thickness and therefore lacks efficiency at high X-ray energies. Consequently, the number of detected photons further needs to be corrected by $eff_{det}\left(E_{fluo}\right)$. In addition, potential detector deadtime τ needs to be addressed in case the detector is exposed to a high photon flux. In summary, the tracer mass in the irradiated area can be calculated with:(1)$$m=\frac{A\;T\left(I_{0}\right)}{I_{0}}\frac{4\pi }{\Omega_{\det\;\;}eff_{det}\left(E_{fluo}\right)\;\left(1-\tau \right)\;}\frac{S}{T\left(E_{fluo}\right)\;\sigma \;\;t}\;.$$

The fluorescence photon number *S* is determined by fitting with the *curve_fit* method from

SciPy in python. The total fit consists of a signal function, a sum of all Gaussian peaks of any elements assumed to be present in the probe, and a background function. The data are fitted in a range of interest around the peaks. Element-specific fluorescence data such as relative line intensity and mean line energy is obtained from the xraylib database [36].

Single line intensities are corrected for energy-dependent detector efficiency and transmission through air or other absorbing material, such that one global free parameter per element is to be fitted: the corrected element amplitude. The mostly flat background is modeled with a linear function and an additional exponential term.

A single-sided hypothesis test is performed to assure that the assumed signal is a valid signal and not only a background fluctuation; hence, the significance is calculated. If the significance exceeds 3 standard deviations, the peak is recognized as a signal. This analysis is performed in an interval of ±3 rms around the mean energy of the peaks. The signal counts *S* are then calculated by integration of the fitted Gaussian from -infinity to infinity.

#### 2.7.2. Fit Routine and Calibration in Single-Cell Measurements

For the quantification of elements, X-ray spectra were peak fitted using a program package based on PyMCA, in order to extract the net intensities of fluorescence lines. The net peak intensities were normalized to the intensity of the incoming monochromatic beam. A multi-element standard reference material (RF-200-0205-C10-X, AXO Dresden GmbH, Dresden, Germany) was employed for external standardization. Element areal densities were calculated from the normalized net peak intensities via comparison with the reference.

Quantitative analysis to determine the cell size and mean intensity values in the elemental maps was carried out with ImageJ [37]. Cell-to-cell uptake variation was determined by fitting a linear function to the reconstructed mass per cell vs. projected cell area and calculating the rms value of the fit-subtracted masses.

## 3. Results

### 3.1. Biological Pre-Assessment of I_2_-Q_10_

Synthesized and labeled I_2_-Q_10_ was qualified and compared to native, non-labeled Q_10_ by testing both human skin cell viability and bioavailability. For this purpose, primary human keratinocytes obtained from three different donors were treated with 10–50 µM I_2_-Q_10_ for 24 h, before cellular metabolic activity (Figure 2a) and cellular enzyme activity were measured (Figure 2b) to ensure non-toxicity of the cells. As standard cell culture Q_10_ concentrations of 50 µM [38] show no negative side effects on cells, a similar concentration can also be used for I_2_-Q_10_. Therefore, this concentration was chosen for further analyses, including bioavailability assessment via liquid chromatography—tandem mass spectrometry (LC-MS/MS). Our results revealed a cellular uptake for I_2_-Q_10_ (Figure 2c) which was comparable to native unlabeled standard Q_10_ (103 ng I_2_-Q_10_/mg total protein content vs. 101 ng Q_10_/mg total protein content).

### 3.2. Quantitative I_2_-Q_10_ Uptake Measurements of Human Skin Cell Pellets

On basis of these positive analytical and biological pre-assessment results, I_2_-Q_10_ uptake was further evaluated by X-ray fluorescence imaging. Cell pellets from keratinocytes obtained from three donors, treated with I_2_-Q_10_ and centrifuged, were filled into glass capillaries and scanned at the P21.1 beamline at the PETRA III synchrotron at DESY, Hamburg, Germany, with an incident energy of 53.1 keV. The goal of these first measurements was the determination of cellular uptake variations between each of the single treated donors, as well as a comparison with untreated skin cells. A summary of the main scan and sample parameters, as well as the resulting iodine mass per cell, is shown in Table 1.

For each capillary, the total iodine mass as well as the iodine distribution in the capillary was reconstructed, as described by Schmutzler and colleagues [2] and in the Methods section. Pixel-by-pixel full spectral analysis consisting of fits to the characteristic iodine K shell fluorescence lines, as well as X-ray fluorescence lines from the detector (Sn, Ba) and background contributions were applied to generate elemental concentration maps. Figure 3a–c shows the two-dimensional iodine distribution maps for the three different donors and d) depicts the map for the untreated skin cell sample from donor 1. The total reconstructed iodine masses in the capillaries were 26.7 ± 4.0 ng, 28.8 ± 4.3 ng and 27.4 ± 4.1 ng for donor 1 (600,000 cells), 2 (600,000 cells) and 3 (400,000 cells), respectively, whereas no iodine could be detected in the untreated cell sample (750,000 cells). In the capillaries from all three donors, iodine was strongly localized at the bottom part of the capillary where the pellet with a height of about 1.5 mm was located. The approx. 9 mm high PBS fluid column above the pellet did not show iodine signals, which was a strong indicator that the cells were still intact during the measurements, without any sign of iodine leakage. Dividing the reconstructed iodine masses by the number of cells per pellet resulted in 44.5, 48 and 68.5 fg iodine/cell for donors 1-3, respectively, in ascending order. In XRF, only control groups containing iodine will yield signals. Since the natural iodine concentration in skin cells was below the detection limit of our XRF experiment, as we could not measure any signals in the untreated skin sample from donor 1 and measurements of the used solutions did not show any iodine impurities, no further control groups were needed. 

In order to address the question of iodine leakage during the incubation process, skin cells obtained from the same donors were treated with iodine only, without being labeled to Q_10_. Two different concentrations were chosen for the treatment, 50 μM and 8.5 μM, of which the first one equals the concentration that was used for the I_2_-Q_10_ treatment and the second one takes the I_2_-Q_10_ NMR purity of 83% into account, meaning, in the worst case, a maximum of 17% pure iodine residue (17% of 50 μM = 8.5 μM). Figure 4 shows the reconstructed iodine masses for one cell sample treated with 50 μM (a) and one treated with 8.5 μM (b), whereby 1 million cells were contained within each pellet. The treatment with 50 μM iodine resulted in a total reconstructed mass of 9.6 ± 1.4 ng which equals 9.6 fg/cell, roughly a factor of five lower than for I_2_-Q_10_ treatment, while in the case of 8.5 μM, no iodine signals were found. This finding is a strong indicator that the uptake is mainly driven by Q_10_-specific and Q_10_-related processes of the cells, and that iodine can be regarded as a well-suited labeling element.

### 3.3. Single Cell Measurements to Determine I_2_-Q_10_ Uptake and Distribution

Having demonstrated the I_2_-Q_10_ accumulation within cell pellets using glass capillaries, a follow-up set of measurements was performed on the single-cell level. As the spatial resolution in XRF is determined solely by the diameter of the X-ray beam applied, a different beamline was chosen for those studies: the Hard X-ray Micro/Nano-Probe Beamline P06 at PETRA III, DESY, Hamburg, Germany. This beamline offers beam diameters in the sub-μm range, a high photon flux, as well as different incident energies up to 40 keV. A summary of the main parameters used for the measurements is given in Table 2.

Human keratinocytes from the same donors as those in the previous experiments were chemically fixated on silicon nitride membrane windows (Silson Ltd., Warwickshire, England) after a 24 h incubation with 50 μM I_2_-Q_10_. Interesting scan areas were determined via high-resolution microscope images of the membranes mounted on a custom-made holder. At the P06 beamline, a membrane with keratinocytes from donor 1 was scanned using a beam with an incident energy of 12 keV, which was chosen due to the optimized background in the iodine L-shell fluorescence region and the high flux achieved in the small beam spot. Figure 5a shows the reconstructed elemental image for iodine resulting from full spectral analysis of all individual pixels in a scan area of 420 × 420 μm^2^ scanned with a resolution of 2 × 2 μm^2^. Panel B) shows the elemental map for three selected cells from the full area shown in A) in an area of 70 × 70 μm^2^, scanned in a subsequent run with higher resolution of 300 × 300 nm^2^.

Iodine is distributed homogenously inside the cells, where Q_10_ is mainly expected, since highest Q_10_ levels were described for the cell organelles mitochondria, lysosomes and Golgi vesicles [39]. Quantitative analysis of the absolute elemental iodine mass per cell for cells with a size between 100 and 350 μm^2^ yielded a mean value of 46.5 ± 10.70 fg/cell, which is in good agreement with the results from the cell pellet measurements shown in Figure 3. Over the full range of all 100 cells, a mean value of 77.03 ± 21.48 fg iodine/cell (≙ 2.6 × 10^−4^ ± 0.73 ng Q_10_/cell) was calculated, with a median value of 49.13 fg iodine/cell (≙ 1.66 × 10^−4^ ng Q_10_/cell). In order to emphasize the homogeneous distribution of iodine and hence Q_10_, the elemental maps of iodine and zinc are compared in Figure 6. The zinc distribution shown in cyan depicted a strong localization in the cell nucleus, different to the homogeneous iodine map colored in magenta. It has to be noted here that the feature of cell nucleus localization is automatically provided in XRF imaging as long as the incident energy is above the K-absorption edge of those elements which are also present in the cell nucleus. Thus, no additional staining is needed, which often alters the kinetics of uptake and distribution and subsequently influences the applicability of final probes [40]. The comparison of the elemental images of zinc and iodine (representing Q_10_) for the fine scan shown in Figure 6c,d once more demonstrates the homogeneous distribution of iodine among the whole cell area, where Q_10_ is primarily expected. The concentration map of zinc can be used to define the outline of cell nuclei and to determine the influence of sample preparation on elemental content in mammalian cells, as discussed in [41].

In order to determine the cell-to-cell uptake variation, the total mass of XRF-marker per cell was reconstructed for all the 100 measured cells and plotted against the projected cell area in the range of 100 to 350 μm^2^, containing the majority of data points. The linear fit function in Figure 7 represents a clear correlation of increasing mass per cell, with the projected cell area determined by ImageJ [37].

## 4. Discussion

The results presented in our work clearly demonstrate the feasibility of XRF imaging to quantitatively determine the uptake and subcellular distribution of iodine-labeled Q_10_. One basic prerequisite of our imaging modality is the development of a labeling strategy, resulting in cellular stability of an element for visualization by XRF imaging. Iodine was found to fulfill the necessary requirements for several reasons: iodine can be bound to Q_10_ without any observed modification of the coenzyme, and obviously there is no detectable natural iodine background in skin cells (Figure 3); thus, there are no overlapping signals with I_2_-Q_10_ treatment. Our reconstructed iodine mass per cell is in good agreement with the expectation value from standard Q_10_ uptake measurements (Figure 2). Control measurements using free iodine, experiments of I_2_-Q_10_ bioavailability via LC-MS/MS as well as the scans around the cell pellets clearly indicate that the measured iodine masses can only be related to the I_2_-Q_10_. In addition, there is neither an indication that the iodine marker was disassembled from the Q_10_ molecule during incubation, nor any noticeable leakage out of the cells.

Our data prove, for the first time, that exogenously applied Q_10_ is taken up by primary human skin cells. In addition, we were able to quantify the Q_10_ uptake and visualize spatial distribution per individual skin cell. The large accumulation of Q_10_, as measured by iodine-labeled Q_10_ signals, within skin cells is in good agreement with previous work, demonstrating that Q_10_ supplementation leads to intracellular effects. In this regard, Prahl et al. [20], have shown ex vivo data in mitochondrial membrane potential stabilization (as a parameter of energetic capacity) in isolated keratinocytes following topical Q_10_ application. Additionally, Schniertshauer et al. [17] demonstrated an increase in mitochondrial respiration, and thus ATP production, after ex vivo treatment of human epidermis samples. Furthermore, inhibition of endogenous Q_10_ biosynthesis in skin cells resulted in a shift towards a senescent phenotype in vitro, demonstrating that Q_10_ depletion is not simply a consequence of skin ageing, but also the main cause of it [42]. Experimentally induced senescence phenotypes in these approaches were reverted by Q_10_ supplementation, replenishing cellular and mitochondrial Q_10_ content. In addition, molecular markers of senescence (β-galactosidase activity, p21 mRNA and protein expression) were rescued, and a significant upregulation of the gene expression of the structural components of the extracellular matrix, namely elastin and collagen type I, alpha I [33] was observed. Application of Q_10_ to aged skin in vivo has been shown to improve the phenotypic signs of ageing by acting both as a restorer of mitochondrial function and as an antioxidant [15,17,19,20,43].

XRF provides new insights into the mode of action of Q_10_ and the beneficial effects of Q_10_ supplementation reported in the scientific literature. According to our findings reported here, the underlying mechanism of the above-mentioned beneficial effects for human skin and skin cells are mainly facilitated by uptake and insertion of exogenous Q_10_ to mitochondrial and cellular membranes, confirming the pivotal role of Q_10_ in regulation and restoration of cutaneous metabolism.

It is noteworthy that iodine can also be used as a labeling element for studies using bigger cellular compositions in three-dimensional contexts, such as skin models. K-shell XRF photons have sufficient energy to penetrate through tissues; thus, using higher-incident X-ray energies with the same marker as demonstrated in this work might be applicable for skin penetration studies as well.

Additionally, the cutting-edge technology of XRF may offer great potential for medical applications as well, reaching from new scientific findings to reducing drug development process efforts, risks and costs. The overall efficiency of the process could be improved by providing a selection of promising candidates and offering insights into the mode of action early on [44]. Thus, exploiting functional and molecular imaging modalities may serve to address and determine pharmacokinetics, drug metabolism, safety aspects and prediction of in vivo efficacy already in the early stages of (pre)clinical studies.

Putative translation from the single-cell level to bigger cellular arrangements such as organ cultures and/or even humans in the future poses challenges for an advanced in vivo XRF imaging approach, as discussed in our previous works [8,9].

## Figures and Tables

**Figure 1 antioxidants-11-01532-f001:**
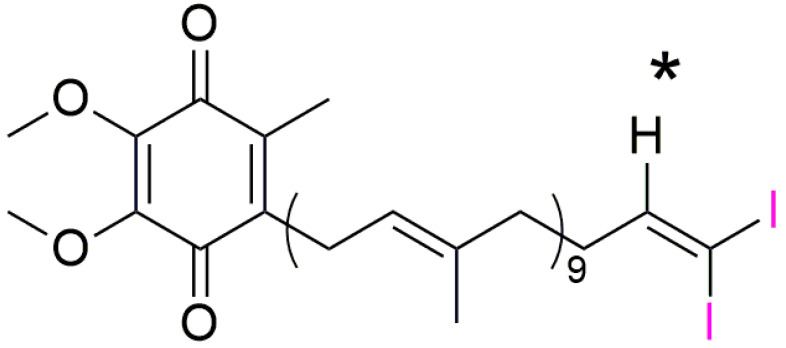
Molecular structure of I_2_-Q_10_ showing the chemical shift of the prominent methine group marked with an asterisk next to the introduced iodine-moiety (marked in pink).

**Figure 2 antioxidants-11-01532-f002:**
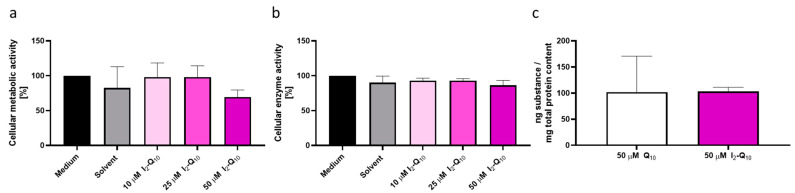
I_2_-Q_10_ pre-assessment. Both (**a**) cellular metabolic activity (MTT assay) and (**b**) cellular enzyme activity (FDA assay) were measured to assess cell vitality at different I_2_-Q_10_ concentrations (*n* = 3, mean ± SD). Bioavailability comparison for (**c**) Q_10_ (*n* = 5 donors, each *n* = 1 technical replicate) and) I_2_-Q_10_ (*n* = 1 donor, *n* = 6 technical replicates); 50 µM treatment, LC-MS/MS, mean ± SD.

**Figure 3 antioxidants-11-01532-f003:**
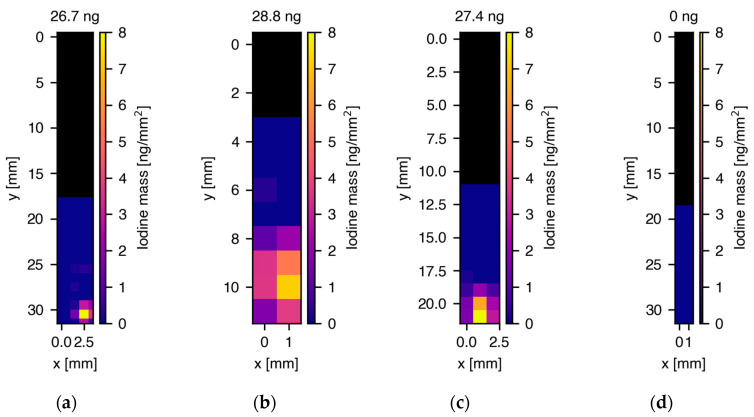
Two-dimensional maps of the spatial distribution and reconstructed masses of iodine in glass capillaries containing cell pellets treated with 50 μM I_2_-Q_10_. (**a**) 26.7 ng total iodine mass in cell pellet from donor 1, (**b**) 28.8 ng total iodine mass in cell pellet from donor 2, (**c**) 27.4 ng total iodine mass in cell pellet from donor 3, (**d**) no iodine signal in untreated cell sample. The maps in (**a**–**c**) show a strong localization of iodine in the bottom part of the capillaries where the cell pellet was located in a PBS fluid column. Above the fluid column, the capillaries were empty, indicated by the black color in the XRF maps. Note the different y-axis scales arising from the varying length and thickness of the hand-blown capillaries.

**Figure 4 antioxidants-11-01532-f004:**
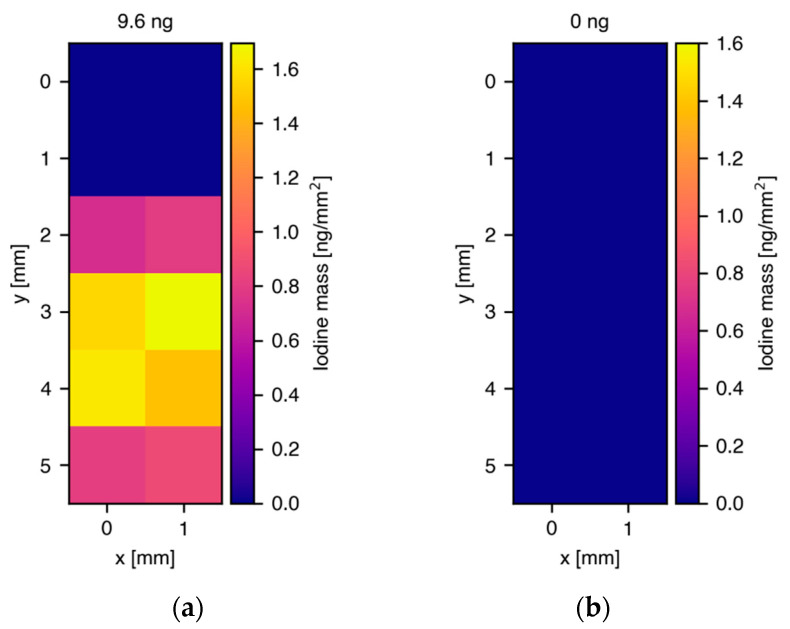
Two-dimensional maps of the spatial distribution and reconstructed masses of iodine in glass capillaries containing cell pellets treated with iodine of different concentrations. (**a**) 9.6 ng total iodine mass in cell pellet from donor 2 after treatment with 50 μM iodine, (**b**) no iodine signal in cell pellet from donor 2 after treatment with 8.5 μ M iodine. Compared with the results for I_2_-Q_10_ treatment, the localization was not as strong, indicating a rather low free iodine uptake and/or background signal from iodine attached to cell surfaces that distributes in the cell-surrounding PBS. Note that in this measurement, a smaller region around the cell pellet was scanned, which was different to results presented in Figure 3, where the scan areas were larger.

**Figure 5 antioxidants-11-01532-f005:**
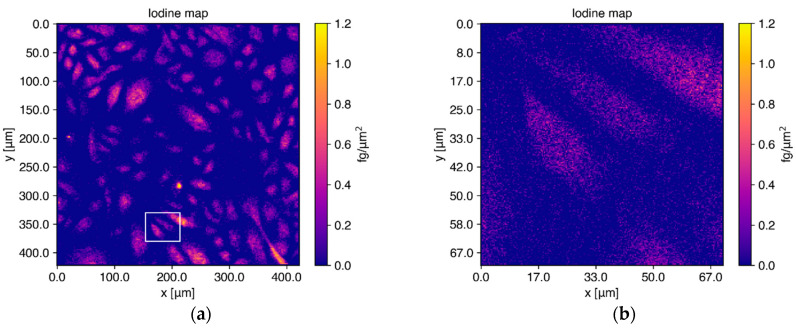
Elemental maps for iodine reconstructed from single-cell measurements after I_2_-Q_10_ treatment. (**a**) Cells from donor 1 on a silicon nitride membrane were scanned with a 12 keV X-ray beam over an area of 420 × 420 μm^2^. Iodine (as labeled to Q_10_) is distributed homogeneously inside the cells. (**b**) Elemental maps of 3 cells from the same membrane scanned again with higher resolution, showing the distributions found in (**a**) in more detail. The scan area is indicated with a white rectangle in the elemental map shown in (**a**).

**Figure 6 antioxidants-11-01532-f006:**
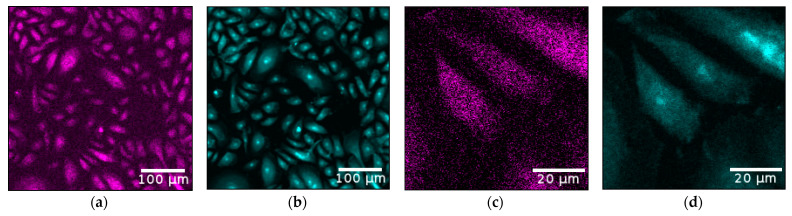
Images of the zinc (cyan) and iodine (magenta) elemental maps for the full scan (**a**,**b**) and the fine scan of 3 selected cells (**c**,**d**), indicated by the white rectangle in Figure 5a. As zinc is mainly located in the cell nucleus, its elemental map is ideally suited to indicate the locations of the nuclei. The direct comparison of iodine and zinc once again emphasizes the homogeneous distribution of iodine among the cytoplasm and agrees with the expected Q_10_ localization.

**Figure 7 antioxidants-11-01532-f007:**
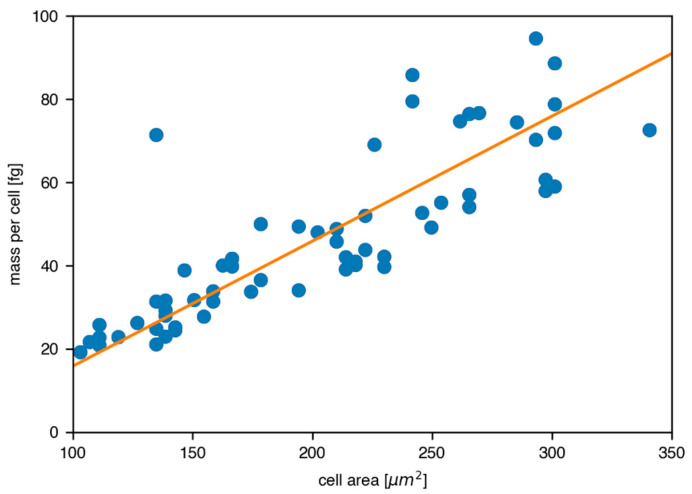
Correlation between reconstructed iodine mass per cell and projected cell area. The orange fit function shows a clear trend of uptake increase with area, and the cell-to-cell uptake variation is 10.70 fg iodine/cell, while the mean value was found to read 46.5 fg iodine/cell.

**Table 1 antioxidants-11-01532-t001:** Summary of the main scan and sample parameters for glass capillary measurements containing keratinocyte cell pellets after different treatment at the P21.1 beamline.

Cell Pellet	Cell Number	Scan Duration [s]	Photons per Pixel	Resolution [mm]	Iodine/Cell [fg]
Donor 1, 50 μM I_2_-Q_10_	600,000	10	1.7 × 10^12^	0.5	44.5
Donor 2, 50 μM I_2_-Q_10_	600,000	10	1.7 × 10^12^	0.5	48
Donor 3, 50 μM I_2_-Q_10_	400,000	10	1.7 × 10^12^	0.5	68.5
Donor 1, no treatment	750,000	10	1.7 × 10^12^	0.5	0
Donor 1, 50 μM Iodine	1,000,000	10	7.5 × 10^11^	1	9.1
Donor 2, 50 μM Iodine	1,000,000	10	7.5 × 10^11^	1	9.6
Donor 2, 8.5 μM Iodine	1,000,000	10	7.5 × 10^11^	1	0

**Table 2 antioxidants-11-01532-t002:** Summary of the main beam parameters used for the measurements at the P06 beamline at PETRA III, DESY, Hamburg. The number sets refer to two different X-ray beam conditions (with and without pre-focusing compound refracting lenses) for the measurements shown in Figure 5 and Figure 6.

	P06, DESY, Hamburg	
Incident energy [keV]	12	
	High-resolution condition	High-flux condition
Photon flux [photons/sec]	1.6 × 10^10^	6 × 10^11^
Beam size [μm^2^]	0.33 × 0.24	2.40 × 2.40

## Data Availability

Data is contained within the article.

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
