# Peer review of "Assessing Cellular Uptake of Exogenous Coenzyme Q10 into Human Skin Cells by X-ray Fluorescence Imaging"

_antioxidants, 2022, doi:10.3390/antiox11081532_

Round 1

Reviewer 1 Report

The article “Assessing Cellular Uptake of Exogenous Coenzyme Q10 into Human Skin Cells by X-Ray Fluorescence Imaging” by Theresa Staufer et al. is dedicated to Q10, a highly conserved coenzyme with antioxidant and bioenergetic properties, is the subject of this investigation to determine its cellular uptake using X-ray fluorescence imaging. The article is a complete and logically structured story of current advances in using X-Ray Fluorescence Imaging to investigate Q10 cellular uptake.  In my opinion, the article should be published in the MDPI Journal of Antioxidants, but a specific revision of the article is needed as outlined below:

1. page 3, line 126. Maybe the percentage of used FBS should be mentioned 

2. page 3, line 134. If instructions for making the I2-Q10 stock solution were supplied, that would be beneficial.

3. page 6, line 259. why Conc. Of 25 µM for I2-q10 is not taken into account for further research when it has been demonstrated to produce a higher percentage of cellular enzyme activity and should require further explanation

4. page 12, figure 6. scale bar ought to have been offered.

Reviewer 2 Report

In manuscript titled “Assessing Cellular Uptake of Exogeneous Coenzyme Q10 into Human Skin Cells by X-Ray Fluorescence Imaging” by Staufer et al., the authors present new technique, X-ray fluorescence (XRF) imaging, for biodistribution studies. The manuscript is multidisciplinary and well-written, however, I have doubts whether the research subject fits the journal. This manuscript presents a new method rather than the actual results of biological research on antioxidants. Role, toxicity and biodistribution of CoQ10 has been excessively studied previously, i.e.:

Knott A, Achterberg V, Smuda C, Mielke H, Sperling G, Dunckelmann K, Vogelsang A, Krüger A, Schwengler H, Behtash M, Kristof S, Diekmann H, Eisenberg T, Berroth A, Hildebrand J, Siegner R, Winnefeld M, Teuber F, Fey S, Möbius J, Retzer D, Burkhardt T, Lüttke J, Blatt T. Topical treatment with coenzyme Q10-containing formulas improves skin's Q10 level and provides antioxidative effects. Biofactors. 2015 Nov-Dec;41(6):383-90. doi: 10.1002/biof.1239. Epub 2015 Dec 9. PMID: 26648450; PMCID: PMC4737275.

Muta-Takada K, Terada T, Yamanishi H, Ashida Y, Inomata S, Nishiyama T, Amano S. Coenzyme Q10 protects against oxidative stress-induced cell death and enhances the synthesis of basement membrane components in dermal and epidermal cells. Biofactors. 2009 Sep-Oct;35(5):435-41. doi: 10.1002/biof.56. PMID: 19753652.

Setoguchi S, Nagata-Akaho N, Goto S, Yamakawa H, Watase D, Terada K, Koga M, Matsunaga K, Karube Y, Takata J. Evaluation of photostability and phototoxicity of esterified derivatives of ubiquinol-10 and their application as prodrugs of reduced coenzyme Q10 for topical administration. Biofactors. 2020 Nov;46(6):983-994. doi: 10.1002/biof.1678. Epub 2020 Oct 6. PMID: 33025665.

Sguizzato M, Mariani P, Spinozzi F, Benedusi M, Cervellati F, Cortesi R, Drechsler M, Prieux R, Valacchi G, Esposito E. Ethosomes for Coenzyme Q10 Cutaneous Administration: From Design to 3D Skin Tissue Evaluation. Antioxidants (Basel). 2020 Jun 3;9(6):485. doi: 10.3390/antiox9060485. PMID: 32503293; PMCID: PMC7346166.

Lohan SB, Bauersachs S, Ahlberg S, Baisaeng N, Keck CM, Müller RH, Witte E, Wolk K, Hackbarth S, Röder B, Lademann J, Meinke MC. Ultra-small lipid nanoparticles promote the penetration of coenzyme Q10 in skin cells and counteract oxidative stress. Eur J Pharm Biopharm. 2015 Jan;89:201-7. doi: 10.1016/j.ejpb.2014.12.008. Epub 2014 Dec 11. PMID: 25500282.

Please find my additional comments below:

1.      Line 106: Why the purity limit of the I2-Q10 is 83%? Does it not affect the quality of the obtained results, especially cell culture experiments?

2.      Line 134: the cited study does not mention the preparation of the stock solution of standard Q10

3.      Line 138: why the mtt method was modified by the authors?

4.      Line 144: “Cellular (I2-) Q10 content was quantified from keratinocytes treated with 50 µM (I2-) Q10 for 24 h as previously described [15]” – The authors of cited study [15] used in their experiments with keratinocytes ubiquinone at a concentration of 18 uM (Fig.5). Why did the authors choose the concentration of 50 uM for bioavailability studies? Moreover, the control of standard Q10 is missing in all cell culture studies (i.e. cell viability studies).

5.      Line 277: “(c) Q10 (n=5 donors, each n=1 technical replicate)  and) I2-Q10 (n=1 donor, n=6 technical replicates); 50 µM treatment, LC-MS/MS, mean ± SD.” Does n = 6 technical replicates mean that the same biological sample was determined 6 times? If so, the authors cannot perform any statistics and compare anything, because it is still the same repetition and only errors / deviations of procedure itself are excluded.

6.      Figure 2: what statistical analysis has been applied?

7.      Table 1: The authors claim that “The goal of these first measurements was the determination of cellular uptake variations between each of the single treated donors as well as a comparison with untreated skin cells. (line 284-285)” Why were the conditions of the experiment not standardized? Controls (no treatment, concentrations, etc) are missing. I.e. only donor 1 represents group “no treatment”. Hence, there is no control of individual groups. Moreover, the number of cells in individual samples is also different - what does this result from?

8.      Lines: 341-347 “2D-maps of the spatial distribution and reconstructed masses of iodine in glass capillaries 341 containing cell pellets treated with iodine of different concentrations. a) 9.6 ng total iodine mass in 342 cell pellet from donor 1 after treatment with 50 μM iodine, b) no iodine signal in cell pellet from 343 donor 1 after treatment with 8.5 μM iodine”. However, table 1 shows that only donor 2 has been treated with 2 different concentrations of iodine.  Why individual experiments were not carried out for 3 samples? Why were different samples selected for different types of experiments?

9.      Line: 382 “Iodine is distributed homogenously inside the cells, where Q10 is mainly expected, 382 since highest Q10 levels were described for the cell organelles mitochondria, lysosomes 383 and Golgi vesicles [37].” However, Figure 5 suggests rather that iodine is cumulated in the cell nucleus. Fig 6 was supposed to dispel doubts about the location, but it is still not clear cut. Verification would be possible by compiling photos of the same area of the zink itself and of the iodine itself, but these data are missing. The accumulation of bright points in the cells in Fig. 6 is the nucleoli, the cell nucleus itself is much larger. 2 signals are visible in this area. However, the verification of my assumptions and the authors' suggestions may only be possible with the presentation of photos of individual signals.

Round 2

Reviewer 2 Report

Thank you for your replies.